# Finding Authentic Counterhate Arguments:
# A Case Study with Public Figures

**Abdullah Albanyan**,[θ] **Ahmed Hassan**,[υ] and **Eduardo Blanco**[γ]

[θ]College of Computer Engineering and Sciences, Prince Sattam Bin Abdulaziz University
[υ]Department of Communication and Information Engineering, Zewail City University
[γ]Department of Computer Science, University of Arizona

a.albanyan@psau.edu.sa   s-ahmed_hassan@zewailcity.edu.eg   eduardoblanco@arizona.edu

## Abstract

We explore authentic counterhate arguments for online hateful content toward individuals. Previous efforts are limited to counterhate to fight against hateful content toward groups. Thus, we present a corpus of 54,816 hateful tweet-paragraph pairs, where the paragraphs are candidate counterhate arguments. The counterhate arguments are retrieved from 2,500 online articles from multiple sources. We propose a methodology that assures the authenticity of the counter argument and its specificity to the individual of interest. We show that finding arguments in online articles is an efficient alternative to counterhate generation approaches that may hallucinate unsupported arguments. We also present linguistic insights on the language used in counterhate arguments. Experimental results show promising results. It is more challenging, however, to identify counterhate arguments for hateful content toward individuals not included in the training set.

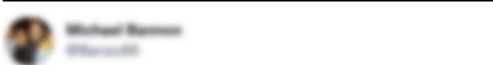

Messi is a racist!!!! Hope he gets suspended. #c▓▓t

---

Synthetic counterhate argument (generic, generated on demand by experts or automatically):
This kind of unsubstantiated statements are not allowed as it demeans and insults others.

---

Authentic counterhate argument from *Dailypost.ng*:

www.dailypost.ng/2012/05/11/[...]-fire-back-drenthes-claims

"The player [Messi] has always shown a maximum respect and sportmanship towards his rivals, something which has been recognized by his [...]"

---

Authentic counterhate argument from *Quora.com*:

www.quora.com/Is-Messi-racist

He [Messi] could be harsh and that's due to the frustration during the game [...], it's all love from Messi.

Figure 1: Hateful tweet (top) and three replies with counterhate. Previous work targets synthetic counterhate arguments that tend to be generic. In this paper, we find authentic counterhate arguments that address the hateful claims in the tweet at hand.

## 1 Introduction

Social media platforms such as Twitter and Facebook are used by many adults—Auxier and Anderson (2021) report that 72% of U.S. adults use at least one social media site. Content in social media can spread fast and help community organizations. For example, the first report of the 2017 Las Vegas shooting was a tweet posted ten minutes after the event started (Blankenship and Graham, 2020). More recently, Twitter and hashtags have been credited with facilitating the organization of protests and the social movement that followed the killing of George Floyd (Wirtschafter, 2021).

Social media has several positive effects including the ability to share information at low cost and being a vehicle for free speech where anyone can express their views. It also has negative effects including the spread of misinformation and hate. Hateful content in social media is widespread. In 2017, a study conducted on 4,248 U.S. adults showed that (a) 41% were personally subjected to online hate speech and (b) 66% witnessed hate speech directed toward others (Duggan, 2020).

Hateful content in social media has consequences in the real world. For example, there is evidence that several celebrities have committed suicide after being the target of hateful content in social media (Today, 2020). Additionally, an analysis of the Japanese daily death registry and one million tweets between 2010 and 2014 revealed that there is a correlation between reports of celebrity suicides in social media and suicides committed by regular people (Ueda et al., 2017).

There have been many efforts to identify hateful content online (Section 2). Social media platforms

have reported spending billion per year on these efforts. Regardless of how hate is identified, one could keep the content but flag it, delete the content, or ban the author. An alternative compatible with freedom of speech is to counter the hateful content, a strategy shown to be effective at minimizing the spread and limiting the consequences of hateful content (Gagliardone et al., 2015). Consider the hateful tweet and three counterhate arguments in Figure 1. Previous work, discussed in Section 2, has targeted synthetic counterhate arguments that (a) are generated on demand by experts (Chung et al., 2019; Fanton et al., 2021) or (b) condemn the language but do not address the specific hateful claims and as a result generate generic counterhate arguments (e.g., "Use of this language is not tolerated and it is uncalled for," (Qian et al., 2019)). In this paper, we instead target authentic counterhate arguments that address the specific hateful claims, as exemplified in the bottom two arguments in Figure 1. We are inspired by previous work outside of the counterhate domain pointing out that (a) URLs increase perceived trust (ODonovan et al., 2012; Morris et al., 2012) and (b) good arguments have to appeal to logic by including factual, testimonial, or statistical evidence (Habernal and Gurevych, 2016). Therefore, authentic counterhate arguments with sources have the potential to be more effective than generic statements condemning hate.

The work presented here works with hateful tweets toward public figures and authentic counterhate arguments published online. A hateful tweet is, according to the Twitter guidelines,[1] any implicit or explicit tweet that attacks an individual's gender, religion, race, ideology, or social class. Our definition of *authentic counterhate argument* borrows from previous work on counterhate (Mathew et al., 2019) and argumentation (Habernal and Gurevych, 2016). We define counterhate as *a direct response that counters hate speech* and an authentic argument as a *paragraph that appeals to logic by including factual, testimonial, or statistical evidence.* In the remainder of this paper, we may refer to *authentic counterhate argument* as *counterhate* to save space. The work presented here could be understood as a specialized information retrieval task where the goal is to identify authentic counterhate arguments against a hateful tweet in (a) online articles and (b) paragraphs within the articles.

The main contributions are:[2]

- a corpus of 250 hateful tweets toward 50 public figures (5 each) and 2,500 candidate articles for authentic counterhate arguments;
- annotations indicating whether the 54,816 paragraphs in the 2,500 candidate articles are authentic counterhate arguments;
- analysis characterizing the language used to make authentic counterhate arguments;
- experimental results showing that the task can be partially automated; and
- qualitative analysis describing when the task is the hardest to automate.

## 2 Previous Work

Identifying hateful content in user-generated content has received substantial attention in recent years (Fortuna and Nunes, 2018). Researchers have presented several datasets and models for hate detection in Twitter (Waseem and Hovy, 2016; Davidson et al., 2017), Yahoo (Warner and Hirschberg, 2012; Djuric et al., 2015; Nobata et al., 2016), Facebook (Kumar et al., 2018), Gab (Mathew et al., 2021), and Reddit (Qian et al., 2019).

Previous efforts have also worked on identifying the target of hate (e.g., a group, an individual, an event or object). For example, Basile et al. (2019) differentiate between groups and individuals, and between immigrant women and other groups. Zampieri et al. (2019a), differentiate between groups, individuals, and others (e.g., an event or object). Similarly, Ousidhoum et al. (2019) detect hateful content in several languages and identify the target from a set of 15 groups.

**Countering Hate** Previous work on countering hate can be divided into two groups: detection and generation. Counter hate detection consists in identifying whether a piece of text counters hateful content. For example, Mathew et al. (2020) identifies counterhate in the replies to hateful tweets. They rely in a simple pattern to detect hateful tweets: *I hate <group>*. Garland et al. (2020) work with German tweets from two well-known groups and define hate and counterhate based on the group the authors belong to. He et al. (2021) work with tweets related to COVID-19 and identify hate and counterhate using 42 keywords and hashtags.

Generating counterhate is arguably more challenging than detecting it. Qian et al. (2019) crowd-

---

[1]https://bit.ly/3J9FpDP

[2]https://github.com/albanyan/counterhate_paragraph

source a collection of counter hate interventions and present models to generate them. Similarly, Fanton et al. (2021) direct their focus to other groups including Jews, LGBTQ+, and migrants. Chung et al. (2019) present CONAN, a collection of anti-Muslim hateful content and counterhate replies. Both the hateful content and counterhate replies in CONAN were written by expert operators on demand; thus, it is unclear whether this synthetic dataset would transfer to content written by regular people. More related to our work, Chung et al. (2021) use external knowledge and GPT-2 to generate counterhate arguments against specific hateful claims, similar to what we define here as *authentic counterhate*. Generating counterhate is certainly a valid strategy, but GPT-2 and other large pretrained models are known for hallucinating facts. In this paper, we bypass this issue and instead retrieve authentic counterhate arguments from online articles. We find counterhate arguments for 72% of hateful content we consider, making this strategy viable. Further, all previous efforts are limited to counterhate to fight against hateful content toward groups. Unlike them, we explore counterhate for hateful content toward individuals.

## 3 A Collection of Hateful Tweets and Authentic Counterhate Arguments

We start our study creating a corpus of hateful tweets and authentic counterhate arguments against the hateful content. We focus on hateful tweets toward 50 public figures which we will refer to as *individuals*. The supplementary materials list the 50 individuals; we collected their names from online sources.[3] The list of 50 individuals is not gender-balanced (male: 30, female: 20) but includes people working in professions such as politician, comedian, singer, and athlete. It also includes, for example, politicians with opposing views including Joe Biden and Donald Trump. Appendix A details the 50 individuals.

### 3.1 Collecting Hateful Tweets toward Individuals

We automatically collect hateful tweets toward the 50 individuals as follows. First, we identify hateful tweets that mention the name of each individual. Second, we impose some filters on these tweets to ensure that the hateful content targets one of the 50

individuals. The second step is necessary to ensure that we do not have hateful tweets that do not target the individual at hand. For example, *You are all a bunch of fu\*\*ers. X is the greatest actor ever* is hateful but it is not hateful toward *X*.

**Selecting Hateful Tweets** First, we retrieve tweets containing the name of any of the 50 individuals using the Tweepy Python library (Roesslein, 2020). Second, we feed these tweets to a hate classifier, HateXPlain model (Mathew et al., 2021), to automatically identify the tweets that are hateful.

**Identifying Hate Segments toward Individuals** Hateful tweets mentioning an individual are not necessarily hateful toward that individual. In order to ensure that the tweets we work with target one of the 50 individuals, we define several patterns grounded on part-of-speech tags. The first pattern consists of the name of the individual followed by *is* and a phrase headed by either a noun or an adjective (e.g., *X is a big a\*\*\*ole*, *X is dumb as rocks*). Appendix B provides the part-of-speech tags we consider for phrases; they are a combination of determiners, adjectives, adverbs, and either a noun or an adjective. The second pattern is inspired by Silva et al. (2016) and consists of *I*, an optional adverb, a hateful verb, and the name of the individual. The list of hateful verbs includes *hate*, *despise*, *detest*, and *abhor* among many others.

We use *segment* to refer to the text matching a pattern. Despite segments come from hateful tweets mentioning an individual, there is no guarantee that the segment is hateful toward the individual or even hateful at all. In order to avoid this issue, we (a) run HateXPlain on the segment and (b) keep *hateful segments* and discard the rest. Here is an example of a hateful tweet mentioning an individual that we discard because there is no hateful segment toward the individual (underlining indicates the segment): *Y'all are an ignorant piece of s\*\*t. X is fu\*\*ing gorgeous* (segment is offensive but not hateful toward X).

We use the part-of-speech tagger in spaCy (Neumann et al., 2019). Table 1 presents the number of individuals and average number of hateful tweets per profession. Politicians and comedians receive more hate (8.6 and 8.0 on average), and hosts and journalists the least (1.0). The average number of hateful tweets is 4.72 across all professions. Based on this average and to ensure we work with hateful tweets targeting a variety of individuals, we re-

[3]https://www.thetoptens.com/hated-people-living-2021/, https://bit.ly/3O5zceh, https://bit.ly/3tk6Uos

| Profession | #Individuals | Avg. #hateful tweets |
|---|---|---|
| Politician | 15 | 8.60 |
| Comedian | 1 | 8.00 |
| Actor | 9 | 3.56 |
| Singer | 11 | 3.54 |
| Athlete | 5 | 3.40 |
| Entrepeneur | 1 | 3.00 |
| Model | 2 | 2.00 |
| Host | 5 | 1.00 |
| Journalist | 1 | 1.00 |
| All | 50 | 4.72 |

Table 1: Number of individuals and average number of hateful tweets per profession of the 50 public figures we work with. The final collection process resulted in 5 hateful tweets per individual.

| | Authentic Counterhate? | | | |
|---|---|---|---|---|
| | no | | yes | |
| Paragraph | 52,451 | (95.7%) | 2,365 | (4.3%) |
| Article | 1,961 | (78.4%) | 539 | (21.6%) |

Table 2: Label percentages at the paragraph and article levels. We work with 54,816 paragraphs and 2,500 articles. The distribution is biased toward no at both levels (i.e., authentic counterhate arguments are rare).

peat the process to collect hateful tweets described above until we obtain 5 tweets containing *unique* hateful segments toward each of the 50 individuals. By *unique*, we mean up to one segment matching a noun or an adjective in the first pattern (*X is [. . . ] noun_or_adjective*) and one segment matching the second pattern (*I optional_adverb hateful_verb X*) per individual. The total number of tweets (and unique segments) is thus 250.

We manually validated the results of the filtering with a sample of 200 discarded hateful tweets and discovered that most of the tweets (83%) were not hateful toward the individual. Here are two more examples: *X is not currently running for political office, dirty haters* (no segment) and *Just so you know: X hates racists* (segment is not hateful).

### 3.2 Collecting Candidates for Authentic Counterhate Arguments

Finding authentic counterhate arguments countering hateful content is harder than it may seem. While finding positive content about anyone with a public presence is generally easy (e.g., *X will save our country from ruin*), finding an argument that directly counters the hateful segment is more nuanced. We design a two-step procedure to first find candidate arguments and then validate them.

**Retrieving Candidate Articles**   For each of the 250 unique hateful segments, we retrieve 10 online articles. We use the following search queries:
- "X is not adjective / noun" for the first pattern, *X is [. . . ] adjective_or_noun*.
- "Reasons to like X" for the second pattern, *I optional_adverb hate_verb X*.

We use these queries with Google Search API and retrieve (a) the top-5 online articles from Quora and (b) the top-5 online articles from other sources. We put an emphasis on Quora articles after we conducted a manual examination and discovered that Quora often has civil conversations where people present their opposing views about a public figure.

**Extracting Text and Paragraphs**   We extract the text from Quora articles using a customized parser built using BeautifulSoup.[4] For other articles, we use Goose,[5] a Python library that abstracts away the parsing details required to extract text from html documents. We then split text into paragraphs using the \<p\> html tag. The result is 54,816 paragraphs from 2,500 articles (50 individuals × 5 hateful tweets × 10 articles = 2,500).

### 3.3 Validating Candidates for Authentic Counterhate Arguments

The last step to collect hateful tweets and authentic counterhate arguments is to manually validate the candidate arguments. We used Label Studio[6] as an annotation tool. The tool showed the hateful tweet with the hateful segment highlighted, and it guided annotators through each paragraph in an article. Annotators decided whether the paragraph was a counterhate argument following the provided annotation guidelines. These guidelines gave enough details on when to label a paragraph as a counterhate paragraph. For example, a paragraph is annotated as counterhate when it criticizes the hateful claim or provides a fact that contradicts the hateful claim. Two annotators participated in the annotation task; both are active social media users and have worked before in the hate and counterhate domain. They invested 180 hours in the annotations. In the first phase of annotations, both of them annotated all paragraphs from ten candidate articles for

---

[4]https://www.crummy.com/software/BeautifulSoup/
[5]https://github.com/grangier/python-goose
[6]https://github.com/heartexlabs/label-studio

| Hateful Tweet: Angelina Jolie is a horrible person with no acting ability |
| :--- |
| *Counterhate paragraph* (yes label): To those people who load nasty comments about celebrities, [...]. I would challenge any of you to do what Jolie does; be a mother, actor, director and a humanitarian while dealing with your own severe health issues. Her work as an ambassador for United Nations has created awareness of the struggles of both children and adult refugees. |
| *Not Counterhate paragraph* (and negative; no label): I have no idea, I am a TV critic & know she shouldn't be considered a A list actor, I believe she has that label because of her dad Being Jon Voit [...]. |
| *Not Counterhate paragraph* (and positive; no label): She is beautiful and has an amazing appearance. |

Table 3: Hateful tweet and three paragraphs from our corpus. The first one contains a counterhate argument. Note that we consider as counterhate arguments statements that counter the specific hateful claims (underlined). The second and third ones are negative and positive toward Angelina Jolie but neither one counters the hateful tweet.

a hateful segment toward each individual (500 articles; 20% of them). The inter-annotator agreement (Cohen's $\kappa$) was 0.75, which indicates *substantial* agreement; coefficients between 0.60 and 0.80 are considered *substantial* agreement and above 0.80 (almost) perfect (Artstein and Poesio, 2008). Given the high agreement, we split the remaining articles in half and each annotator annotated one half (all paragraphs in 1,000 articles each).

## 4 Analyzing Counterhate Arguments

Table 2 shows the label percentages (no and yes) at the paragraph and article levels. Out of the 54,816 paragraphs in the 2,500 online articles, only 4.3% are authentic counterhate arguments. In other words, authentic counterhate arguments are rare despite we designed the retrieval of online articles to improve the chances of finding counterhate. The number of articles having at least one paragraph containing an authentic counterhate argument is higher (21.6%). Despite the low percentages, we note that our collection includes at least one paragraph containing an authentic counterhate argument for 72% of hateful tweets. This is true despite limiting ourselves to finding authentic counterhate arguments in 10 online articles per hateful tweet. We also note that it is more likely to find counterhate arguments for certain professions. For example, the total number of counterhate arguments is low for professions such as journalists and hosts (averages are 4 per individual and 6.4 per individual, respectively). On the other hand, the number is high for professions such as athletes and politicians (averages are 135.8 per individual and 46.1 per individual, respectively). Additionally, regarding the number of counterhate arguments per individual, we find that the lowest and the highest number of counterhate arguments per individual are

2 and 264, respectively, while the average number of counterhate arguments per individual is 47.4.

We present a hateful tweet and three paragraphs from our corpus in Table 3. The tweet is a hateful tweet toward *Angelina Jolie* (i.e., a horrible person). Annotators chose yes for the first paragraph as it presents reasons not to believe that *Angelina Jolie is a horrible person*, the hateful segment in the tweet. On the other hand, the next two paragraphs are not authentic counterhate. The second paragraph is negative toward Angelina Jolie and criticizes her acting abilities—it doesn't address the hateful segment. The third paragraph is positive toward Angelina Jolie but does not counter the hateful tweet with a convincing argument— being beautiful and having good appearance are arguably not incompatible with being a *horrible person*.

### 4.1 Linguistic Analysis

We conduct a linguistic analysis comparing the language in (a) paragraphs that contain authentic counterhate arguments (yes) and those who do not (no); and (b) articles that have at least one paragraph containing an authentic counterhate argument and those who do not (Table 4). We tokenize and part-of-speech tag text to count pronouns and proper nouns using spaCy (Neumann et al., 2019). In order to identify negation cues, we use a RoBERTa-based cue detector trained with CD-SCO (Morante and Blanco, 2012). We use a profanity lexicon to identify profanity words,[7] and consider words misspelled if they do not appear in the Brown corpus (Francis and Kucera, 1979) or the lexicon of profanity words. Finally, we use a sentiment lexicon containing positive and negative words (Mohammad and Turney, 2013) and TextBlob[8] to ob-

---

[7]https://bit.ly/3OpdYYP
[8]https://github.com/sloria/TextBlob

| | Paragraph | Article |
|---|---|---|
| Number of . . . | | |
| tokens | | ↑↑↑ |
| pronouns | ↑↑↑ | ↑↑↑ |
| first person | | ↑↑ |
| second person | ↓↓↓ | |
| proper nouns | ↑↑↑ | |
| negation cues | ↑↑↑ | ↑↑↑ |
| question marks | ↓↓↓ | |
| profanity words | ↓↓↓ | ↓↓ |
| misspellings | ↓↓↓ | |
| positive words | ↑↑↑ | ↑↑↑ |
| negative words | ↓↓↓ | |
| Polarity score | ↓↓↓ | |
| Subjectivity score | ↑↑↑ | ↑↑↑ |

Table 4: Linguistic analysis of counterhate arguments at the paragraph and article levels. Arrow direction indicates whether higher values indicates counterhate (up) or not (down). Number of arrows indicate the p-value (t-test; one: $p < 0.05$, two: $p < 0.01$, and three: $p < 0.001$). All tests pass the Bonferroni correction.

tain polarity and subjectivity scores. We make several interesting observations:

- Longer articles are more likely to contain counterhate, but paragraph length is not a good indicator of counterhate.
- The more pronouns the more likely is a paragraph or article to contain counterhate. The more second-person pronouns in a paragraph, however, the less chances of being counterhate. This is because these pronouns (*you*, *your*) are often used to attack the author of the hate rather than the hateful content.
- Negation cues, positive words, and subjective language indicate counterhate in both paragraphs and articles.
- Profanity, misspellings, and negative words are rare in paragraphs containing counterhate.

## 5 Experiments and Results

We experiment with classifiers to determine (a) whether a paragraph is an authentic counterhate argument against a hateful tweet and (b) whether an article contains a paragraph that is an authentic counterhate argument against a hateful tweet. All the neural classifiers are built using pretrained language models from HuggingFace (Wolf et al., 2020) and Pytorch (Paszke et al., 2019). We tune hyperparameters with the training and development splits, and report results with the test split. We first report results with a random 70/10/20 split. Then, we explore a more realistic—and challenging—scenario: using in the test set candidate counterhate arguments for hateful tweets toward *unseen individuals during training* (Section 5.2).

**Baselines** We also present results with two baselines: majority and random labels. The majority label is no for both paragraphs and articles.

**Paragraph-Level Neural Classifier** We experiment with neural classifiers built on top of a BERT-based transformer. Specifically, we use the RoBERTa transformer (Liu et al., 2019), which is pretrained with 800M words from the BooksCorpus (Zhu et al., 2015), English Wikipedia (2,500M words), CC-News (63M English news articles) (Nagel, 2016), OpenWebText (Gokaslan et al., 2019), and Stories (Trinh and Le, 2018). The neural architecture consists of RoBERTa, a fully connected layer with 128 neurons and ReLU activation, and another fully connected layer with 2 neurons and a softmax activation which outputs the prediction (*no* or *yes*). We tried several input choices (individually and combinations):

- the article title,
- the hateful tweet,
- the hateful segment, and
- the candidate paragraph.

In order to feed to the network with different combinations of inputs, we concatenate them with the separator special token .

**Article-Level Neural Classifier** We experiment with the same architecture as the paragraph-level classifier, however, we use the Longformer transformer (Beltagy et al., 2020) instead of RoBERTa. The Longformer can handle longer input texts. We also consider several inputs:

- the article alone,
- the hateful tweet, and
- the hateful segment.

We also experiment with a system that aggregates the paragraph-level predictions. This system reuses the neural classifier at the paragraph level and outputs yes if any of the paragraphs are identified to be an authentic counterhate argument.

### 5.1 Quantitative Results

First, we present experimental results with a random 70/10/20 split (Table 5). Second, we present results using 70/10/20 split in which there is no overlap between individuals (toward whom the hate

|  | P | R | F1 |
|---|---|---|---|
| **Paragraph-Level Predictions** | | | |
| Majority baseline | 0.00 | 0.00 | 0.00 |
| Random baseline | 0.04 | 0.50 | 0.08 |
| RoBERTa trained with . . . | | | |
| article title | 0.54 | 0.19 | 0.28 |
| hateful tweet | 0.00 | 0.00 | 0.00 |
| + article title | 0.59 | 0.22 | 0.32 |
| hateful segment | 0.44 | 0.06 | 0.10 |
| + article title | 0.61 | 0.21 | 0.32 |
| paragraph | 0.62 | 0.46 | 0.53 |
| + article title | 0.57 | 0.76 | 0.65 |
| + hateful tweet | 0.65 | 0.76 | 0.70 |
| + article title | 0.64 | 0.77 | 0.70 |
| + hateful segment | 0.68 | 0.75 | 0.71 |
| + article title | 0.67 | 0.76 | 0.71 |
| pretraining | 0.69 | 0.75 | 0.72 |
| **Article-Level Predictions** | | | |
| Majority baseline | 0.00 | 0.00 | 0.00 |
| Random baseline | 0.21 | 0.50 | 0.30 |
| LongFormer trained with . . . | | | |
| article | 0.70 | 0.24 | 0.36 |
| + hateful tweet | 0.62 | 0.30 | 0.41 |
| + hateful segment | 0.67 | 0.29 | 0.41 |
| Aggregate par.-level preds. | 0.65 | 0.85 | 0.74 |

Table 5: Results for the yes label. We present results for identifying counterhate arguments in paragraphs and articles using several inputs. *Aggregate par*agraph-*level pred*ictions predicts yes if any of the paragraph-level predictions for an article is yes. The appendix provides additional results with more input combinations and pretraining with more related tasks.

is directed) in the training, development, and test splits (Section 5.2). The results in the paper are for the most important label: yes (i.e., identifying authentic counterhate arguments). The appendices present detailed results including Precision, Recall, and F1-measure for no and yes labels.

**Paragraph-Level Counterhate** Table 5 presents the results at the paragraph level (top block), i.e., identifying whether a paragraph is an authentic counterhate argument for a given hateful tweet. The article title alone outperforms the baselines but obtains modest results (F1: 0.28). Either the hateful tweet or hateful segment by itself yields poor results (F1: 0.00, 0.10), leading to the conclusion that the hateful content (both the full tweet or only the segment) is not a good signal for whether one is more likely to find a counterhate argument.

The paragraph alone obtains much better results (0.53), signaling that counterhate arguments are

somewhat identifiable despite it is unknown what hate they are countering. Surprisingly, combining the paragraph with the article title yields much better results (F1: 0.65). Recall stays roughly constant for any combination of two or more inputs that include the paragraph (R: 0.75–0.77). Increasing precision, however, requires the hateful tweet or segment in the input. Either one yields roughly the same results (P: 0.57 vs. 0.65 and 0.68; F1: 0.70 and 0.71). These results lead to the conclusion that it is unnecessary to identify the hateful segment prior to feeding the tweet to the network.

We also explore pretraining with complementary tasks, but it is barely beneficial. Our rationale is to leverage existing datasets with counterhate examples, which are the minority class in our corpus (Table 2). We adopt the method by Shnarch et al. (2018), which incorporates annotated data from related tasks using different ratios in each training epoch. In the first epoch, all instances from the related task are used for training, and the ratio is decreased all the way to zero in the last epoch. The corpora we pretrain with are CONAN (Chung et al., 2019) and Multitarget-CONAN (Fanton et al., 2021). CONAN consists of 6,654 hate speech examples (related to Islamophobia) paired with synthetic counterhate statements written by experts. Multitarget-CONAN is similar but includes hateful content toward multiple targets. It includes 5,003 examples collected with a semi-automatic procedure. Pretraining is barely beneficial (F1: 0.72 vs. 0.71). We hypothesize that it is due to two facts. First, CONAN and Multitarget-CONAN include hateful content toward groups (Muslims, LGBTQ+, etc.) rather than individuals. Second, the counterhate in these corpora is synthetic and does not address the hateful claims with what we refer to as an argument; it condemns the hateful content without arguing against it.

**Article-Level Counterhate** The results at the article level (Table 5, bottom block) present similar trends. The Longformer using the text in the article alone outperforms the baseline (F1: 0.36), and including either the hateful tweet or just the segment brings the F1 to 0.41. In other words, while the model can identify good arguments just by looking at the article, it is beneficial to know the hateful content the counterhate argument is for.

We obtain the best results (F1: 0.74 vs.0.41) aggregating the paragraph-level predictions. In this setup, we output yes at the article level if any of

| | P | R | F1 |
|---|---|---|---|
| Paragraph-Level Predictions | | | |
| Random baseline | 0.03 | 0.53 | 0.06 |
| RoBERTa trained with . . . | | | |
| paragraph | 0.44 | 0.30 | 0.36 |
| + article title | 0.42 | 0.36 | 0.39 |
| + hateful tweet | 0.49 | 0.32 | 0.39 |
| + article title | 0.33 | 0.47 | 0.39 |
| + hateful segment | 0.45 | 0.38 | 0.41 |
| + article title | 0.41 | 0.43 | 0.42 |
| pretraining | 0.43 | 0.41 | 0.42 |

Table 6: Results for the yes label testing with candidate counterhate arguments for hateful tweets toward public figures not seen during training. We present results with the best performing systems from Table 5. Results are much lower (F1: 0.72 vs. 0.42), showing that the task is challenging for unseen individuals.

| Error Type | % |
|---|---|
| Undesirable counterhate | 33 |
| Positive but not countering hateful claims | 15 |
| Unsupported counterhate statement | 10 |
| Ambiguous, mix of hate and counterhate | 8 |
| Intricate text | 32 |
| Implicit counterhate argument | 14 |
| Mix of named entities | 18 |
| General world knowledge | 11 |
| Rhetorical question | 5 |

Table 7: Error types made by the best performing model (RoBERTa trained with *paragraph+ hateful segment + article title* and pretrained with related tasks, Table 5). We provide the percentages for each error type.

the paragraphs are predicted yes by the paragraph-level classifier. These result lead to the conclusion that the Longformer faces challenges finding counterhate paragraphs in long articles, as many articles have many paragraphs (over 21 on average) and only few, if any, contain authentic counterhate.

## 5.2 Is it Harder to Find Counterhate Arguments for Unseen Individuals?

Yes, it is (Table 6). The results in terms of relative improvements depending on the combination of inputs show similar trends, but all of them are substantially lower (best F1: 0.42 vs. 0.72). Despite making sure to have unique hateful segments in our corpus (Section 3), these results show that learning to identify authentic counterhate arguments for a new individual is challenging. We hypothesize that the reason is that the same counterhate argument can be reused to counter against several hateful claims toward the same individual.

## 5.3 Qualitative Analysis

While our best model obtains somewhat high results at the paragraph level (F1: 0.72; first block in Table 5), we conduct a qualitative analysis to describe when it makes the most errors. Table 7 lists the error types and their frequency in a random sample of 100 errors made by the best model. These error types overlap (i.e., a paragraph may contain many named entities and a rhetorical question).

The most common error types (33%) fall under *undesirable counterhate argument* including:

- Positive comments about the individual not countering the hateful claims:
  - Hateful tweet: *Messi is a racist!!! [...]*
  - Paragraph: *Messi is one of the best soccer players [...] Messi's goals makes him the highest goalscorer.* Gold: no, Pred.: yes.
- Unsupported counterhate that is not a counterhate argument according to our definition:
  - Hateful tweet: *Pelosi is an EVIL B**CH!!!!!*
  - Paragraph: *She isn't.* Gold: no, Pred. : yes.
- Mix of hate and counterhate arguments:
  - Hateful tweet: *Demi Lovato is a fat b**ch and I hate her like there is no tomorrow.*
  - Paragraph: *I think at certain times in her life she has been very attractive [...]. But then there have been times when she has been somewhat over weight.* Gold: no, Pred.: yes.

The second most common error types fall under intricate text (32%):

- Implicit counterhate arguments:
  - Hateful tweet: *Reminder that Mel Gibson is a Jew-hating racist skid-mark on [...].*
  - Paragraph: *Mel has worked in Hollywood for years and Hollywood has a prominent Jewish community. If I remember correctly Mel also had Jewish friends that defended him after the drunk driving incident.* Gold: yes, Pred.: no
- Mix of named entities:
  - Hateful tweet: *This is why I dislike Bush and lost a lot of respect for the man.*
  - Paragraph: *No. Ronald Reagan was the best President [...] I cannot comment on any presidents prior to 1970.* Gold: no, Pred.: yes.

Lastly, we identify two additional error types:

- Counterhate that requires world knowledge:
  - Hateful tweet: *Bill Gates is an evil [...]*
  - Paragraph: *people don't know facts about*

*bill gates, and are instead believing in false conspiracy theories.* Gold: yes, Pred.: no.

- Rhetorical questions:
  - Hateful tweet: *Ronaldo is wayyyyy better [...] monkey Messi is selfish and a dickhead.*
  - Paragraph: *How selfish of him to donate to people who need it!* Gold: yes, Pred.: no.

## 6 Conclusions

Countering hate is effective at minimizing the spread of hateful content (Gagliardone et al., 2015). Unlike blocking content or banning users, countering hate does not interfere with freedom of speech. Previous work (a) works with hateful content toward groups and (b) focuses on generic, synthetic counter arguments written on demand by experts or the crowd. In this paper, we focus on authentic counterhate arguments against hateful claims toward individuals. Authentic arguments, unlike generic ones, address specific hateful claims and include factual, testimonial, or statistical evidence.

We present a collection of 250 hateful tweets toward 50 individuals and candidate counterhate arguments. Candidates come from 54,816 paragraphs from 2,500 online articles. Our annotation effort shows that authentic counterhate arguments are rare (4.3%). Experimental results show promising classification results. It is more challenging, however, to identify authentic counterhate arguments for a hateful claim toward an unseen individual.

## Limitations

The work presented here has several limitations. First, our retrieval approach assumes that an online article with an authentic counterhate argument for a given hateful tweet is available. Empirically, we found this is the case for 72% of hateful tweets despite we limit our retrieval to 10 online articles. That said, there is no guarantee that the approach would work for a brand new tweet.

Another limitation of this study is the restrictions imposed on hateful tweets. We use patterns and classifiers to identify hateful tweets and segments. Neither one is 100% perfect and may introduce biases. Lastly, the results with hateful tweets toward unseen individuals during training show a large drop in results. More robust models are needed in this scenario.

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

# A   Individuals

Table 8 presents the 50 individuals we used in our study. The table shows their names, professions, and genders. The 50 individuals cover various professions such as models, politicians, actors, entrepreneurs, hosts, journalists, comedians, singers, and athletes.

| # | Individual | Profession | Gender | # | Individual | Profession | Gender |
|---|---|---|---|---|---|---|---|
| 1 | Ellen Degeneres | Host | Female | 2 | Sean Hannity | Host | Male |
| 3 | Nancy Grace | Host | Female | 4 | Jay Leno | Host | Male |
| 5 | Tucker Carlson | Host | Male | 6 | Chrissy Teigen | Model | Female |
| 7 | Kim Kardashian | Model | Female | 8 | Bill Gates | Entrepreneur | Male |
| 9 | Bill Cosby | Comedian | Male | 10 | Piers Morgan | Journalist | Male |
| 11 | Jimmy Carter | Politician | Male | 12 | Newt Gingrich | Politician | Male |
| 13 | Justin Trudeau | Politician | Male | 14 | Chris Christie | Politician | Male |
| 15 | Nancy Pelosi | Politician | Female | 16 | David Duke | Politician | Male |
| 17 | Joe Biden | Politician | Male | 18 | Donald Trump | Politician | Male |
| 19 | George Bush | Politician | Male | 20 | Dick Cheney | Politician | Male |
| 21 | Boris Johnson | Politician | Male | 22 | Hillary Clinton | Politician | Female |
| 23 | Barack Obama | Politician | Male | 24 | Bill Clinton | Politician | Male |
| 25 | Mitch McConnell | Politician | Male | 26 | Ann Coulter | Actor | Female |
| 27 | Mel Gibson | Actor | Male | 28 | Lindsay Lohan | Actor | Female |
| 29 | Meghan Markle | Actor | Female | 30 | Angelina Jolie | Actor | Female |
| 31 | Will Smith | Actor | Male | 32 | Demi Lovato | Actor | Female |
| 33 | Amber Heard | Actor | Female | 34 | Sandra Bullock | Actor | Female |
| 35 | Justin Bieber | Singer | Male | 36 | Nicki Minaj | Singer | Female |
| 37 | Jennifer Lopez | Singer | Female | 38 | Madonna | Singer | Female |
| 39 | Chris Brown | Singer | Male | 40 | Beyonce | Singer | Female |
| 41 | Britney Spears | Singer | Female | 42 | Avril Lavigne | Singer | Female |
| 43 | Kanye West | Singer | Male | 44 | Anne Hathaway | Singer | Female |
| 45 | John Mayer | Singer | Male | 46 | LeBron James | Athlete | Male |
| 47 | Floyd Mayweather | Athlete | Male | 48 | Lionel Messi | Athlete | Male |
| 49 | Neymar | Athlete | Male | 50 | Cristiano Ronaldo | Athlete | Male |

Table 8: List of the 50 individuals we work with.

## B  Part-of-Speech Patterns

Table 9 shows the patterns grounded on part-of-speech tags. These patterns are used to identify the hate segments toward individuals (Section 3.1 in the main paper). Patterns 1-21 consist of the name of the individual followed by is and a noun or an adjective phrase. The second kind of patterns, 22 and 23, was inspired by Silva et al. (2016).

## C  Inter-Feature Correlations

Figures 2 and 3 present the linguistic features used in Section 4 of the main paper. We present correlations at the paragraph and article levels. Most correlation coefficients are low, meaning that our linguistic features capture different kinds of language.

## D  Implementation Details

We used the Python's Transformers library to load the base RoBERTa (Liu et al., 2019) and Longformer (Beltagy et al., 2020) models. Our dataset was pre-processed by removing URLs, removing symbols, removing any additional spaces, and at the end, converting all words to lower-case. The pre-processed data is then fed to RoBERTa and Longformer models where RoBERTa and Longformer tokenizers were used respectively to tokenize tweets and to obtain the ids and attention masks. RoBERTa model was used at the paragraph level while the Longformer model was used at the article level. All models were trained using a learning rate of 1e-5, a batch size of 16, an AdamW optimizer (Loshchilov and Hutter, 2017), and a sparse categorical cross-entropy loss function. The RoBERTa model used a maximum sequence length of 256 tokens (padding shorter sequences), whereas the Longformer model used a maximum sequence length of 2048 tokens, as articles are much longer than paragraphs. All models were trained for 6 epochs while saving a checkpoint of the model parameters after the epoch in which the model achieved the lowest validation loss.

## E  Detailed Results

Tables 10 and 11 show the detailed results complementing Tables 5 and 6 in the paper. We provide Precision, Recall, and F1-measure for the yes and no labels for the paragraph and article levels using

| # | Pattern | # | Pattern |
|---|---------|---|---------|
| 1 | [individual's name, is, 'JJ'] | 2 | [individual's name, is, 'JJ', 'NN'] |
| 3 | [individual's name, is, 'NN'] | 4 | [individual's name, is, 'NN', 'NN'] |
| 5 | [individual's name, is, 'DT', 'JJ'] | 6 | [individual's name, is, 'DT', 'JJ', 'NN'] |
| 7 | [individual's name, is, 'DT', 'JJ', 'NN', 'NN'] | 8 | [individual's name, is, 'DT', 'JJS'] |
| 9 | [individual's name, is, 'DT', 'JJS', 'NN'] | 10 | [individual's name, is, 'DT', 'NN'] |
| 11 | [individual's name, is, 'DT', 'NN', 'NN'] | 12 | [individual's name, is, 'DT', 'NN', 'IN', 'NN'] |
| 13 | [individual's name, is, 'DT', 'NNP'] | 14 | [individual's name, is, 'DT', 'NNP', 'VBG'] |
| 15 | [individual's name, is, 'DT', 'VBG', 'NN'] | 16 | [individual's name, is, 'DT', 'RBS', 'JJ'] |
| 17 | [individual's name, is, 'DT', 'RBS', 'VBN'] | 18 | [individual's name, is, 'PDT', 'DT', 'JJ'] |
| 19 | [individual's name, is, 'PDT', 'DT', 'JJ', 'NN'] | 20 | [individual's name, is, 'PDT', 'DT', 'NN'] |
| 21 | [individual's name, is, 'RB', 'JJ'] | 22 | [ I , <hateful verb>, individual's name] |
| 23 | [ I , RB, <hateful verb> , individual's name] | | |

Table 9: Lists of sequences of part-of-speech tags used to identify hate segments. DT: Determiner, NN: Singular noun, JJ: Adjective, JJS: Superlative Adjective, RB: Adverb, RBS: Superlative Adverb, VBG: Present participle verb, VBN: Past participle verb, PRP: Pronoun, and PDT: Predeterminer.

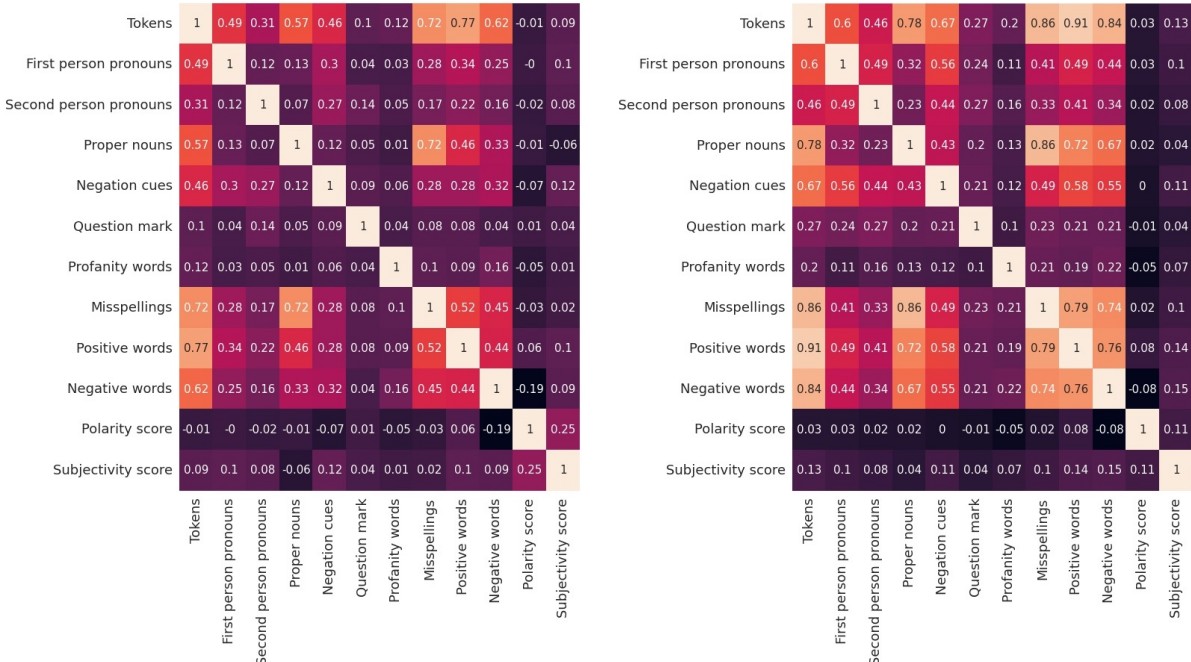

Figure 2: Correlation coefficients between features used in the linguistic analysis. The heatmap shows the correlations with paragraphs that are counterhate (left) and are not counterhate paragraphs (right).

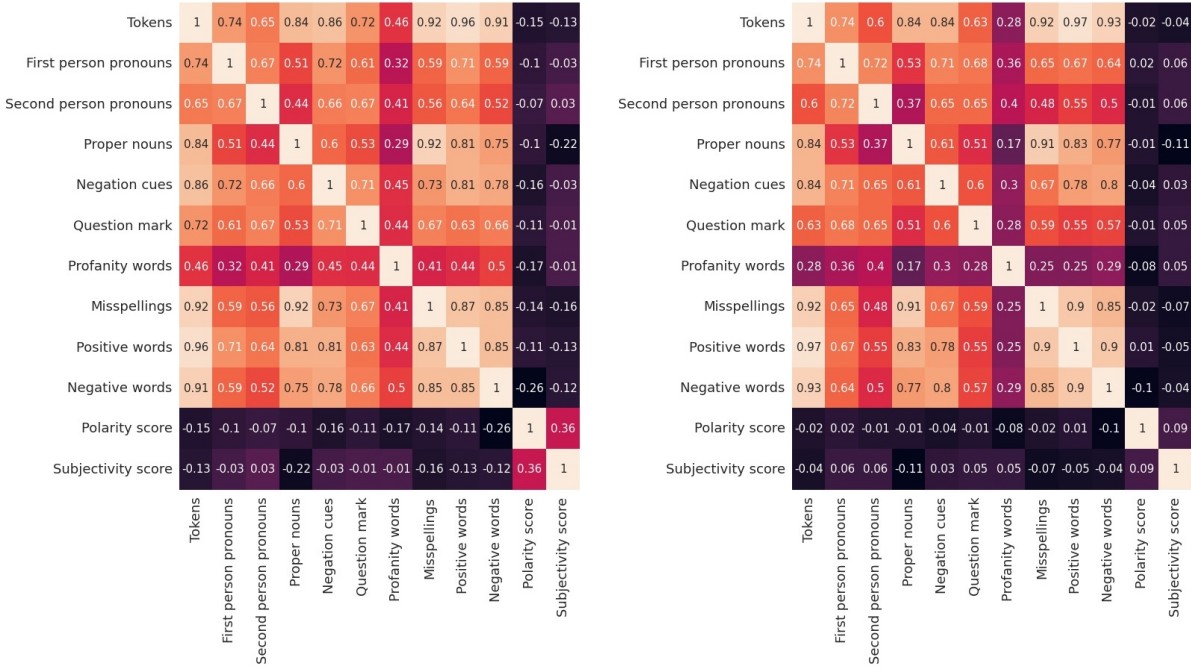

Figure 3: Correlation coefficients between features used in the linguistic analysis. The heatmap shows the correlations with articles that are counterhate (left) and are not counterhate articles (right).

RoBERTa and Longformer.

The top block of table 10 provides additional results for pretraining RoBERTa with more existing corpora:

- hate: tweets containing and not containing hateful content (Basile et al., 2019).
- sentiment: tweets that are positive, neutral, or negative (Rosenthal et al., 2017).
- irony: irony and not irony tweets (Van Hee et al., 2018).
- offensive: offensive or not offensive tweets (Zampieri et al., 2019b).
- CONAN and Multitarget-CONAN were discussed in details in Section 5.1.

| | Yes | | | No | | | Weighted Avg. | | |
|---|---|---|---|---|---|---|---|---|---|
| | P | R | F | P | R | F | P | R | F |
| **Paragraph-Level Predictions** | | | | | | | | | |
| Majority baseline | 0.00 | 0.00 | 0.00 | 0.96 | 1.00 | 0.98 | 0.92 | 0.96 | 0.94 |
| Random baseline | 0.04 | 0.50 | 0.08 | 0.96 | 0.50 | 0.66 | 0.92 | 0.50 | 0.63 |
| RoBERTa trained with … | | | | | | | | | |
|   article title | 0.54 | 0.19 | 0.28 | 0.96 | 0.99 | 0.98 | 0.95 | 0.96 | 0.95 |
|   hateful tweet | 0.00 | 0.00 | 0.00 | 0.96 | 1.00 | 0.98 | 0.92 | 0.96 | 0.94 |
|     + article title | 0.59 | 0.22 | 0.32 | 0.97 | 0.99 | 0.98 | 0.95 | 0.96 | 0.95 |
|   hateful segment | 0.44 | 0.06 | 0.10 | 0.96 | 1.00 | 0.98 | 0.94 | 0.96 | 0.94 |
|     + article title | 0.61 | 0.21 | 0.32 | 0.97 | 0.99 | 0.98 | 0.95 | 0.96 | 0.95 |
|   paragraph | 0.62 | 0.46 | 0.53 | 0.98 | 0.99 | 0.98 | 0.96 | 0.96 | 0.96 |
|     + article title | 0.57 | 0.76 | 0.65 | 0.99 | 0.97 | 0.98 | 0.97 | 0.96 | 0.97 |
|     + hateful tweet | 0.65 | 0.76 | 0.70 | 0.99 | 0.98 | 0.99 | 0.97 | 0.97 | 0.97 |
|       + article title | 0.64 | 0.77 | 0.70 | 0.99 | 0.98 | 0.99 | 0.97 | 0.97 | 0.97 |
|     + hateful segment | 0.68 | 0.75 | 0.71 | 0.99 | 0.98 | 0.99 | 0.97 | 0.97 | 0.97 |
|       + article title | 0.67 | 0.76 | 0.71 | 0.99 | 0.98 | 0.99 | 0.97 | 0.97 | 0.97 |
|         pretrained with … | | | | | | | | | |
|           Hate | 0.62 | 0.78 | 0.69 | 0.99 | 0.98 | 0.98 | 0.97 | 0.97 | 0.97 |
|           Irony | 0.61 | 0.78 | 0.69 | 0.99 | 0.98 | 0.98 | 0.97 | 0.97 | 0.97 |
|           Offensive | 0.67 | 0.73 | 0.70 | 0.99 | 0.98 | 0.99 | 0.97 | 0.97 | 0.97 |
|           Sentiment | 0.64 | 0.78 | 0.70 | 0.99 | 0.98 | 0.99 | 0.97 | 0.97 | 0.97 |
|           CONAN and | | | | | | | | | |
|           Multitarget-CONAN | 0.69 | 0.75 | 0.72 | 0.99 | 0.99 | 0.99 | 0.98 | 0.97 | 0.98 |
| **Article-Level Predictions** | | | | | | | | | |
| Majority baseline | 0.00 | 0.00 | 0.00 | 0.78 | 1.00 | 0.88 | 0.61 | 0.78 | 0.69 |
| Random baseline | 0.21 | 0.50 | 0.30 | 0.77 | 0.48 | 0.60 | 0.65 | 0.49 | 0.53 |
| LongFormer trained with … | | | | | | | | | |
|   article | 0.70 | 0.24 | 0.36 | 0.82 | 0.97 | 0.89 | 0.79 | 0.81 | 0.77 |
|     + hateful tweet | 0.62 | 0.30 | 0.41 | 0.83 | 0.95 | 0.89 | 0.78 | 0.81 | 0.78 |
|     + hateful segment | 0.67 | 0.29 | 0.41 | 0.83 | 0.96 | 0.89 | 0.79 | 0.81 | 0.79 |
| Aggregate. par.-level preds. | 0.65 | 0.85 | 0.74 | 0.96 | 0.87 | 0.91 | 0.89 | 0.87 | 0.87 |

Table 10: Detailed results (P, R, and F) for RoBERTa and Longformer to predict whether a paragraph and an article is *counterhate*. The results complement Table 5 in the main paper.

| | Yes | | | No | | | Weighted Avg. | | |
|---|---|---|---|---|---|---|---|---|---|
| | P | R | F | P | R | F | P | R | F |
| **Paragraph-Level Predictions** | | | | | | | | | |
| Random baseline | 0.03 | 0.53 | 0.06 | 0.97 | 0.50 | 0.66 | 0.95 | 0.50 | 0.64 |
| RoBERTa trained with … | | | | | | | | | |
|   paragraph | 0.44 | 0.30 | 0.36 | 0.98 | 0.99 | 0.99 | 0.97 | 0.97 | 0.97 |
|     + article title | 0.42 | 0.36 | 0.39 | 0.98 | 0.99 | 0.98 | 0.97 | 0.97 | 0.97 |
|     + hateful tweet | 0.49 | 0.32 | 0.39 | 0.98 | 0.99 | 0.99 | 0.97 | 0.97 | 0.97 |
|       + article title | 0.33 | 0.47 | 0.39 | 0.98 | 0.97 | 0.98 | 0.97 | 0.96 | 0.96 |
|     + hateful segment | 0.45 | 0.38 | 0.41 | 0.98 | 0.99 | 0.98 | 0.97 | 0.97 | 0.97 |
|       + article title | 0.41 | 0.43 | 0.42 | 0.98 | 0.98 | 0.98 | 0.97 | 0.97 | 0.97 |
|         pretraining | 0.43 | 0.41 | 0.42 | 0.98 | 0.98 | 0.98 | 0.97 | 0.97 | 0.97 |

Table 11: Detailed results (P, R, and F) for RoBERTa whether a paragraph is *counterhate* for unseen individuals. The results complement Table 6 in the main paper.