# OpenReview forum: "Finding Authentic Counterhate Arguments: A Case Study with Public Figures"
_EMNLP/2023/Conference — EMNLP 2023 Main_

### Official Review · Reviewer_xM6k · 2023-07-20

**Soundness:** 4

**Excitement:**

3: Ambivalent: It has merits (e.g., it reports state-of-the-art results, the idea is nice), but there are key weaknesses (e.g., it describes incremental work), and it can significantly benefit from another round of revision. However, I won't object to accepting it if my co-reviewers champion it.

**Paper Topic And Main Contributions:**

This paper addresses the detection of counter hate arguments with a focus on insults/hate speech targeting public figures. The authors introduce a collection of 250 hateful tweets towards 50 individuals combined with 2,500 online articles (each addressing one of the 50 individuals) in which each paragraph has been labelled as being counter hate or not. The hateful tweets and the additional online articles are obtained using surface patterns and information retrieval. Apart from a brief descriptive linguistic analysis on their novel dataset, the authors present classification experiments using that dataset. They address two tasks: detection of counter hate on the paragraph level and counter hate on the article level. The classifiers the authors examine are based on finetuning transformers (RoBERTa and Longformer). For the paragraph-level classification, the authors also consider the difficult setting in which training and test data use different individuals. The resulting classifiers perform notably worse.

The paper basically makes 3 contributions: It introduces a new data resource, it carries out a computationally-aided linguistic analysis on it and presents classification results (NLP engineering experiment).


**Reasons To Accept:**

The paper is very well-written (no significant typos or grammar mistakes) and has a clear structure. All experiments are well-motivated. The authors are also very specific about their implementation (always mentioning the tools they used along their configuration). The experimental set-up looks mostly sound. The dataset is carefully constructed. The chosen annotators look sufficiently qualified which is also confirmed by the acceptable interannotation agreement. The classifiers that are chosen including the baselines look appropriate for these data. The classification results look fairly plausible. I also think that the results are appropriately discussed and interpreted. I particularly appreciate the brief linguistic analysis in Table 4 which helps the reader get a better idea what type of language counter hate represents. I also think that the experiments on unseen individuals (Section 5.2) is important. It shows that one only gets reasonable classification performance if one has training data for the public figure that one is going to find counter hate for.

**Reasons To Reject:**

Despite the apparent technical soundness of the approach, I really wonder how pressing the given task is. Counter hate regarding identity groups and private citizens seems much more pressing to me. In other words, the negative effects on those groups caused by hate speech seem to be much more severe. It comes with the territory of being a public figure that those individuals are often subject to intensive criticism. This, of course, does not excuse in any way the use of hate speech, insults or defamation directed towards them. My impression is simply that these people are much more prepared of what they face on the web than private individuals which usually are much less resilient. I acknowledge and appreciate that the authors try to give some motivation and also cite Ueda et al. (2017). Still, I find it not that convincing that celebrity suicides also causes an increase of suicide of private individuals. (Regarding Ueda et al. (2017), I am a bit sceptical in how far these results are representative in general and particularly of other cultural areas others than the one addressed in that publication, i.e. Japan.) My pessimistic guess is that given that public figures are often significantly less protected by the law against insults than private citizens, operators of social media sites would be more reluctant to invest in software that specifically combats insults/hate speech regarding public figures.

One important insight of the paper is that the classification approach is only effective when the test data focuses on public figures that have already been observed in the training data. Given this insight, I wonder whether the proposed method could in generally scale up to a fully-fledged commercial application. Would it be worthwhile to have such a tool if one can only effectively act against hate speech directed towards people for whom one has collected specific training data?

Finally, I am a bit worried that the extraction of the hate tweets and the retrieval of online articles might have restricted the scope of possible instances of hate speech and also possible instances of counter speech. Both hate tweets and online articles were retrieved using fairly simple surface-based lexical patterns. I wonder whether this approach might have biased the resulting hate tweets and online articles since the underlying query terms are likely to be lacking in lexical diversity.


**Reproducibility:**

4: Could mostly reproduce the results, but there may be some variation because of sample variance or minor variations in their interpretation of the protocol or method.

**Reviewer Confidence:**

3: Pretty sure, but there's a chance I missed something. Although I have a good feel for this area in general, I did not carefully check the paper's details, e.g., the math, experimental design, or novelty.

**Typos Grammar Style And Presentation Improvements:**

-	The authors use the word “despite” incorrectly. It is a preposition rather than a subordinate conjunction. Please write something like “Despite the fact that …”

-	Lines 300-306, “The tool showed … on the annotation guidelines.”: I found this sentence hard to read, please rephrase.

-	There is something wrong with the line indices of line 497, 612 and 717.

---

> ### Author Rebuttal · Authors · 2023-08-25
>
> Thank you for the detailed review.
>
> **Reasons to Reject**
> - RE Targeting celebrities:
> It is certainly subjective, but we would argue that addressing hate speech targeting celebrities would improve online discourse in general. Doing so could benefit all participants (less hate), not just the celebrities (by minimizing potential consequences the celebrities may suffer).
>
> **RE grammar.**  We will fix the grammar/typos issues. We really appreciate the time you took to point out these typos and grammar issues.

---

### Official Review · Reviewer_pTGV · 2023-07-29

**Soundness:** 3

**Excitement:**

4: Strong: This paper deepens the understanding of some phenomenon or lowers the barriers to an existing research direction.

**Paper Topic And Main Contributions:**

This paper tackles the problem of retrieving relevant counterhate (counterspeech) arguments to counter online hateful content targeting individuals. The authors present a collection of 250 hateful tweets toward 50 individuals plus counterhate arguments retrieved from  2,500 online articles. They also tested various strategies for developing classifiers to identify potential counterhate articles.
As counterhate arguments are rare and existing datasets mostly are synthetic, the dataset (not provided for review) and experiments provided can help characterise counterhate in-the-wild and hate mitigation.


**Questions For The Authors:**

1. Do you have any numbers on how HateXPlain performs on the data? e.g., how many filtered tweets are hateful/not hateful toward each profession/individual?
2. Are there statistics of counterhate arguments found/annotated (as yes) for each profession/individual? e.g., is it easier to find counterhate arguments for certain professions/individuals?

**Reasons To Accept:**

1. A new dataset consists of 250 hateful tweets paired with ~2400 counterhate arguments/paragraphs.
2. Linguistic analysis of authentic counterhate characteristics and error analysis of when the classification task is hard.

**Reasons To Reject:**

1. I appreciate the effort done in all the experiments on the performance of classifiers across different strategies for getting relevant counterhate arguments (section 5.1), but I don't think it adds much new knowledge to what we already knew based on findings from information retrieval.
2. Detail about the annotation guideline is missing

**Reproducibility:**

3: Could reproduce the results with some difficulty. The settings of parameters are underspecified or subjectively determined; the training/evaluation data are not widely available.

**Reviewer Confidence:**

5: Positive that my evaluation is correct. I read the paper very carefully and I am very familiar with related work.

**Typos Grammar Style And Presentation Improvements:**

In the paper, it is not clear whether the annotated counterhate paragraphs are useable directly in response to hateful tweets as they are provided or not. I would suggest adding a few sentences explaining if further editing on the paragraphs is needed to be used as responses.
Also, the detail about the annotation guideline is very little - what is the criteria for a paragraph to be annoataed as counterhate argument? What qualifies as positive statistical evidence (line 307) - when numbers are mentioned? It is better to elaborate on it more.

---

> ### Author Rebuttal · Authors · 2023-08-25
>
> **Annotation guidelines.** More details about the annotation instructions and guidelines will be added in the extra page if accepted. Thank you for pointing this out.
>
> **Answer for Q1**
> We do not know the numbers per profession/individual but in general, we retrieved approximately 41,000 tweets for the 50 individuals. Then, as discussed in L204, we fed these tweets to the HateXPlain classifier. We ended this process with 2,040 hateful tweets (5% of total tweets), approximately 38,960 tweets were not classified as hateful.
>
>
> **Answer for Q2**
> Yes, but it is not included in the current version of the paper. For example, it is more likely to find counterhate arguments for a politician than a journalist. We believe that such insights should and can be included in the corpus analysis section of the paper. Thank you, we will include them using the extra page if accepted.

---

### Official Review · Reviewer_bvb9 · 2023-08-04

**Soundness:** 3

**Excitement:**

4: Strong: This paper deepens the understanding of some phenomenon or lowers the barriers to an existing research direction.

**Missing References:**

- The claim on L067 that previous work only has generic condemnations isn't true. Both Chung et al., 2019 and Mathew et al., 2019 (already cited) include analysis of different hate countering strategies. Furthermore, Allaway et al., 2022 (https://arxiv.org/pdf/2303.16173.pdf) construct counterhate statements that also target specific components of the hatespeech. This should be included in the references.
- There are also non-synthetic arguments considered in Mathew et al., 2019, as well as the works cited on L139-147 (in fact this paragraph describes how they are not synthetic). The task as presented in the paper is very much a classification task, so it does not make sense to only consider generation approaches (which are generating synthetic data) as comparisons for that.

**Paper Topic And Main Contributions:**

The authors construct a corpus of hateful tweets towards individual public figures with related articles. The articles are labeled at the paragraph and article level for whether they contain substantiated (not unsupported) counterhate arguments for a specific hateful segment in the tweet. The tweets are collected by identifying specific syntactic patterns of hate. The accompanying articles are taken from a google search with emphasis placed on responses from Quora. A subset of the paragraphs from articles are labeled by two annotators with strong agreement and the remaining are labeled by a single annotator. The authors conduct an analysis of the language in countering hate and also train classifiers for the task of predicting whether a paragraph or argument is countering hate in a tweet. The results show that the task is quite hard, especially for individuals not seen during training.

Main contributions: new data resource

**Questions For The Authors:**

- A: L201 -- how many tweets are retrieved each time? If all tweets targeting an individual are retrieved the first time, how can this process be repeated (as stated on L246)?
- B: do you validate the hate annotations for the non-discarded tweets in 3.1?
- C: L336 -- Were there any individuals without any counterhate arguments? Were there specific professions that had the most countering?
- D: L458 -- does this mean classifying a paragraph as counter hate in a tweet without seeing either the tweet or the paragraph?
- E: L479 -- did you experiment with pretraining for the hateful tweet + article title combination? If not, why? It seems odd to only have pretraining on the last combination. Do you think the lack of benefit from pretraining could be in part due to the differences in data because the new data has specific patterns of hate which may not be in the other datasets?


**Reasons To Accept:**

- The dataset contains good annotations and will likely be useful for other researchers on identifying and also generating or analyzing counterspeech.
- The focus on individuals is novel and an important area for counterspeech.

**Reasons To Reject:**

- The data was constructed with only specific patterns of hate, which may introduce biases. Although the authors acknowledge this limitation, they should include some analyses of potential biases or more discussion of this.
- The claim that previous work has only considered generic denouncing arguments or synthetic data is not correct. See note in missing references.

**Reproducibility:**

4: Could mostly reproduce the results, but there may be some variation because of sample variance or minor variations in their interpretation of the protocol or method.

**Reviewer Confidence:**

4: Quite sure. I tried to check the important points carefully. It's unlikely, though conceivable, that I missed something that should affect my ratings.

**Typos Grammar Style And Presentation Improvements:**

- The authors should include an ethics section discussing ethical issues related to studying hate specifically targeting individuals. In particular, the data could be used to train a model to produce hate targeted at individuals.
- L099 ("It") and L112 ("task"): the task is really clearly defined before the first reference to it, and although it can be inferred from the rest of the paper, there really should be a clear definition in the introduction.
- L069: "result generic" --> "result has generic"
- L158: I would argue that statements constructed by experts are not synthetic because they are human written.
- L225: "come from" --> "coming from"
- L227: "segment is hateful" --> "segments are hateful"
- L249-253: it is not clear whether this sentence means the tweet has to contain both types of segments or just one. From the example on 261 it seems like just one, but this should be clarified.
- L329: "despite we designed" --> "despite having designed"
- L337: "we limit ourselves" --> "limiting ourselves"
- L341: "Angelina Jolie (horrible person)" -- this phrasing makes it sound like it is your opinion that AJ is a horrible person. I think just remoe the parenthetical.
- L354: this would be easier to read if it was a subsection instead of a paragraph heading.
- L366: "misspellings" --> "misspelled"
- L518: "This results lead to" --> "This result leads to"
- L529: "we took care" --> "taking care"
- There are weird floating line numbers on the last two pages (497 in Table 5, 563 in table 7, 612 in L562, and 529 in L579).

---

> ### Author Rebuttal · Authors · 2023-08-25
>
> **Reasons to Reject**
>
> RE Although the authors acknowledge this limitation, they should include some analyses of potential biases or more discussion of this.
> - We will expand on the discussion in the extra page if accepted. Thank you for the suggestions
>
> RE The claim that previous work has only considered generic denouncing arguments or synthetic data is not correct. See note in missing references.
> - Thank you, we will rephrase and include the additional references.
>
> **Questions for The Authors**
>
> - Answer to Q(A):
> In each iteration of the collection process, we retrieve a batch of tweets per individual (tweets containing the name of any of the 50 individuals) until we collect 5 tweets that contain unique hateful segments toward each individual (using the HateXplain model). Once we have 5 unique hateful segments toward an individual, we remove the individual from the collection process and continue repeating these steps for the remaining individuals. We will clarify: the process is repeated for each individual.
>
> - Answer to Q(B):
> Yes, we manually validated that 1) the tweets are actually hateful, and 2) we have 5 unique hateful segments for each individual.
>
> - Answer to Q(C):
> Some hateful tweets do not have any counterhate arguments; however, each individual has many counterhate arguments (each individual has 5 hateful tweets).
>   - The lowest number of counterhate arguments per individual is 2
>   - The highest number of counterhate arguments per individual is 264
>   - The average number of counterhate arguments per individual is 47.4
>
>   And regarding the number of counterhate arguments per profession, we found the number of counterhate arguments is low for some professions like journalists and hosts while high for professions like politicians and athletes. It is interesting to include this kind of analysis in the final version of the paper. Thank you.
>
> - Answer to Q(D):
> Yes. Let us clarify that this is (probably) the simplest baseline. We include it to show that the problem cannot be reduced to "look at the title and then just tell me whether the article has a counterhate argument.
>
> - Answer to Q(E):
> No, we found that (hateful tweet + article) yields a lower F1-score compared to (hateful tweet + segment), so we decided to do the pretraining with this combination (hateful tweet + segment); additionally, the combination (hateful tweet + segment) obtains the best results compared to all other inputs.
> Regarding the second part of the question, “Do you think the lack of…..“ we believe this is due to the fact that the two datasets that we do the pretraining with (CONAN and Multitarget-CONAN) are towards groups instead of individuals, and their counterhate arguments are synthetic as stated in L498-L504.
>
> **Grammar / Typos**
>
> Thank you. We will fix the grammar/typos issues and include the ethics section if accepted.

---

### Meta-Review · Area_Chair_T6YR · 2023-09-19

**Recommendation:** 3

**Metareview:**

All the reviewers agree on the fact that the paper provides valuable and novel contributions to research in countering hate speech messages online. A corpus of hateful tweets towards individual public figures with related articles is created, and annotated with useful annotations to identify and analyze counterspeech. The experimental set-up looks also solid and sound.
The main concerns (not answered in the rebuttal) are related to the fact that data was constructed with only specific patterns of hate, which may introduce biases, as well as missing details about the annotation instructions and guidelines. A discussion on how far the results provided in the paper are representative in general is also missing.

---

### Decision · Program_Chairs · 2023-10-07

**Decision:**

Accept-Main

**Comment:**

All the reviewers agree on the fact that the paper provides valuable and novel contributions to research in countering hate speech messages online. A corpus of hateful tweets towards individual public figures with related articles is created, and annotated with useful annotations to identify and analyze counterspeech. The experimental set-up looks also solid and sound.
The main concerns (not answered in the rebuttal) are related to the fact that data was constructed with only specific patterns of hate, which may introduce biases, as well as missing details about the annotation instructions and guidelines. A discussion on how far the results provided in the paper are representative in general is also missing.